# Phenotypic Discordance in Siblings with Identical Compound Heterozygous *PARK2* Mutations

**DOI:** 10.3390/brainsci7070071

**Published:** 2017-06-24

**Authors:** David Isaacs, Daniel Claassen, Aaron B. Bowman, Peter Hedera

**Affiliations:** 1Department of Neurology, Vanderbilt University Medical Center, 1301 Medical Center Dr. Suite 3930, Nashville, TN 37232-5400, USA; david.a.isaacs@vanderbilt.edu(D.I.); daniel.claassen@vanderbilt.edu (D.C.); 2Department of Pediatrics, Pediatric Neurology, Vanderbilt University Medical Center, Nashville, TN 37232-2594, USA; aaron.bowman@vanderbilt.edu

**Keywords:** *PARK2*, parkin, Parkinson’s disease, DAT-SPECT

## Abstract

*PARK2* mutations are the most common cause of early-onset Parkinson’s disease. No genotype-phenotype correlation exists, and phenotypic variability is quite common. We report two siblings with confirmed identical compound heterozygous mutations in the *PARK2* gene manifesting strikingly different phenotypes. The older brother demonstrated marked parkinsonism by his mid-20’s, whereas the younger brother developed exercise-induced dystonia in his mid-30’s with no subsequent clinical progression, highlighting the clinical heterogeneity of the disease and implying the role of other genetic and/or environmental factors in disease progression. The younger sibling, despite his mild symptoms, had a clearly abnormal dopamine transporter (DaT)-SPECT scan. To our knowledge, this is the first such reported case of an abnormal DaT-SPECT scan in a patient with biallelic *PARK2* mutations who does not meet the clinical criteria for Parkinson’s disease.

## 1. Introduction

Over the past twenty years, seven monogenic forms of Parkinson’s disease (PD) have been discovered. Four genes (*SNCA*, *LRRK2*, *VPS35*, and *EIF4G1*) are associated with autosomal dominant inheritance patterns and three (*PARK2*, *PINK1*, and *DJ1* (*PARK7*)) with recessive inheritance patterns [1]. *PARK2* mutations (MIM 602544), first identified in Japanese families, are the most common known cause of early-onset parkinsonism worldwide [2,3]. In a series of 246 patients with PD from families with at least one member affected before age 45, 38 (15%) had *PARK2* mutations; prevalence of the mutations declined with older- age-of onset, being present in 67% with onset before 20 but only in 7% with onset after 29 [4,5]. Other studies report lower frequencies of *PARK2* mutations, ranging from 1.4–8.2%, in those with early-onset PD (variably defined as symptom emergence before age 40 or 45) [1]. Patients with older age-of-onset, even into the seventh decade, have been identified [4,5,6].

*PARK2* encodes parkin, a ubiquitin E3 protein ligase that facilitates tagging proteins for proteasomal degradation [7,8]. Pathogenic *PARK2* mutations result in loss-of-function of parkin, which, mounting evidence indicates, leads to neurodegeneration primarily via defective mitophagy [7,8]. Over 100 pathologic *PARK2* mutations have been reported, including copy number variations, missense mutations, and exon rearrangements [5,9,10,11,12]. Pathologically, parkin-related PD is characterized by neuronal loss in the substantia nigra pars compacta and locus coeruleus; Lewy bodies, the distinctive pathologic feature of idiopathic PD, are notably absent in the vast majority of patients [3,13,14]. This, in conjunction with other pathologic distinctions, has led some investigators to postulate that parkin-related PD and idiopathic PD are unique clinico-pathologic entities [3].

No distinct genotype-phenotype correlation exists in patients with parkin-related PD [5]. Nonetheless, as compared to idiopathic PD, patients with biallelic *PARK2* mutations have younger age-of-onset and slower disease progression, as well as greater likelihood of dystonia, motor fluctuations, symmetric motor manifestations, sleep benefit, and hyperreflexia [3,5]. Unlike typical idiopathic PD, freezing, festination, retropulsion, and falls can develop early in parkin-related PD [15]. Significant phenotypic heterogeneity exists, with reports of patients manifesting exercise-induced dystonia, focal dystonia, akinetic-rigid parkinsonism, late-onset parkinsonism indistinguishable from idiopathic PD, and even cerebellar dysfunction [6,15]. Marked intra-familial variability can occur, and even patients with the same *PARK2* mutation can exhibit distinct phenotypes [6,15,16].

Here we report a case of two brothers with confirmed identical compound heterozygote *PARK2* mutations, manifesting strikingly distinct phenotypes, highlighting the clinical heterogeneity and implying the role of other genetic and/or environmental factors.

## 2. Case

At 12 years of age, the proband developed bilateral, non-painful, involuntary curling of his toes, occurring without warning or precipitant. There was no diurnal variation. Around age 18, an intermittent bilateral upper extremity action tremor emerged, with uncertain response to alcohol. By his mid-20s, the patient began to experience significant postural and gait impairment with freezing and retropulsion. He was finally diagnosed with PD in his mid-30’s. Pramipexole provided robust symptomatic benefit initially. Evaluation prior to establishing care at our clinic was notable for a brain magnetic resonance imaging (MRI) without contrast, which was unremarkable, and testing for serum ceruloplasmin and copper levels, which were normal. The patient presented to our clinic at age 40, by which time he was experiencing significant motor fluctuations (peak-dose dyskinesias and random off periods), off-state lower extremity dystonia, and progressively worsening freezing of gait. In terms of non-motor symptoms, he suffered from occasional orthostatic hypotension, sleep fragmentation, and daytime somnolence; saliently, he had no symptoms of rapid eye movement (REM) behavior disorder (RBD), hyposmia, or cognitive impairment. Genetic testing ultimately revealed that the patient was a compound heterozygote for mutations in *PARK2*. One allele had a 40 base pair deletion in Exon 3, spanning from position 337 to 376 resulting in a frameshift (known disease-causing mutation) (Figure 1). The second allele had a deletion spanning exons 5 and 6 (also a known disease-causing mutation).

The patient’s dystonia gradually progressed, with eventual asymmetric involvement of the upper and lower extremities. At age 42, he underwent bilateral subthalamic nucleus (STN) deep brain stimulation (DBS), which provided several years of significant motoric benefit. However, due to STN-stimulation- and medication-refractory dystonia, at age 50 he elected to have additional DBS leads implanted into the bilateral globus pallidus interna (GPi). Unfortunately, the dystonic symptoms only mildly improved with this second DBS intervention. Currently, at 53, the patient is functionally wheelchair-bound. Comprehensive neuropsychological evaluation shortly after bilateral GPi DBS demonstrated mild executive dysfunction, but otherwise intact cognition.

The proband’s younger brother, six years his junior, was asymptomatic until age 36, when he developed progressive cramping and curling in his right toes with running distances of five miles or more. He exercised 1–2 days per week on average. Initial examination in our clinic, at age 39, revealed minimal hypomimia, but no tremor, bradykinesia rigidity, or gait abnormalities. Foot dystonia could not be elicited in clinic. Selegiline resolved his exercise-induced foot dystonia. The patient has no PD non-motor symptoms, other than chronic constipation; specifically, he manifested no autonomic dysfunction, RBD, or hyposmia. His most recent examination, at age 47, was unchanged from his initial clinic encounter; specifically, he did not meet UK Brain Bank Criteria for PD [17]. Genetic testing revealed *PARK2* mutations identical to his brother (Figure 1).

In terms of family history, the brothers have no other siblings. Neither their parents nor their four grandparents displayed any neurologic symptoms. No other family members were known to suffer from any movement disorders. The father was diagnosed with chronic leukemia. The family has no reported history of consanguinity. The proband has two children the younger brother has one child; all are currently asymptomatic. The brothers did not endorse unique environmental exposures to heavy metals or chemicals; both were raised together in the same household. Serum uric acid levels were 0.30 mmol/L in the proband and 0.29 mmol/L in his younger brother. The brothers drank similar quantities of caffeine (1–2 cups of coffee daily during their adult life). Human stem cell-derived neuroprogenitors from both brothers showed the same degree of increased cytotoxicity with copper and cadmium [18].

Both brothers underwent dopamine transporter (DaT)-SPECT scans (DaT scans) (Figure 2). The proband’s DAT scan, at age 49, demonstrated markedly reduced uptake in the bilateral putamina, with relatively intense symmetric activity in the caudate heads. His younger brother’s scan, performed at age 42, was also clearly abnormal, showing asymmetrically decreased activity in the putamen bilaterally.

## 3. Discussion

We report two siblings carrying biallelic mutations in the *PARK2* gene who were phenotypically discordant, with one sibling demonstrating marked parkinsonism by his mid-20’s and the second developing exercise-induced dystonia in his mid-30’s with no subsequent clinical progression. Similar phenotypic discrepancy in patients with identical PARK2 mutations has previously been reported. In their case series, Khan et al. identified two brothers with PD due to identical *PARK2* mutations, one with tremor-predominant disease (Patient 12) and one who has remained atremulous for 47 years (Patient 13) [15]. Deng et al. described a family with five siblings sharing identical compound heterozygous *PARK2* mutations, four of whom developed early-onset PD between the ages of 30 and 38, while the fifth, aged 56 at the time of the publication, remained asymptomatic [16]. Klein et al. studied four siblings with identical compound heterozygous *PARK2* mutations; one became symptomatic at age 31, while another was asymptomatic until age 64 [6].

The reason for this phenotypic heterogeneity remains unclear. Various environmental toxins have been implicated in PD, including pesticides and metals, but in our patients and those referenced above, no clear exposure differences were identified [18,19]. Further, analysis of copper neurotoxicity in early stage neuroprogenitors derived from induced pluripotent stem cells revealed no differences between these brothers in sensitivity of their mitochondria to acute copper exposure, despite increased sensitivity of both relative to control neuroprogenitors [18].

A further interesting aspect of this case report is the clearly abnormal DaT scan in the younger brother, who could be classified as having presymptomatic Parkinson’s disease. Various functional neuroimaging is employed to characterize parkinsonian conditions in vivo, two of the most common studies being [^18^F]fluorodopa PET and DaT scans, which can assess presynaptic dopaminergic deficit [20]. [^18^F]fluorodopa PET studies in parkin-related PD have produced mixed results. One study found no significant difference in striatal uptake between parkin and non-parkin early-onset PD patients [20]. In contrast, another study demonstrated that patients with *PARK2* mutations have more severe and diffuse presynaptic dopaminergic deficits, compared to patients with non-parkin early-onset PD [21].

DaT scans has demonstrated more severe and symmetric loss of striatal signal in patients with parkin-related PD as compared to those with non-parkin-related PD, though notably the parkin-related PD patients had significantly earlier age-of-onset (31 vs. 39 years) and longer disease duration (15 versus 6 years) [13]. The authors did note that the greater symmetry of signal reduction in parkin-related PD is consistent with the greater frequency of symmetric motor symptoms in this population [13]. In our patients, the younger brother, manifesting solely exercise-induced dystonia and minimal hypomimia with no evidence of clinical progression, has a clearly abnormal DaT scan with marked reductions in striatal dopamine transporter expression. This is, to our knowledge, the first report of an abnormal DaT scan in a patient with biallelic *PARK2* mutations not meeting criteria for PD. Notably, patients with non-degenerative causes of dopa-responsive dystonia (e.g., mutation in GTP cyclohydrolase I gene) have normal DaT scans [22]. In a case series of parkin-related PD patients by Khan et al., one patient (Patient 10) had an [^18^F]dopa PET scan that showed more-than-expected presynaptic dopaminergic deficit, relative to a patient with idiopathic PD and similar motor impairment [15]. Interestingly, many asymptomatic patients with a single *PARK2* mutation have significant reduction in striatal [^18^F]dopa uptake [23]. The impact of single heterozygous *PARK2* mutations remains unclear, but prior literature has reported such patients manifesting extrapyramidal symptoms [24]. Investigation is ongoing as to the role of heterozygous *PARK2* mutations in late-onset PD [25].

## 4. Conclusions

This case report highlights phenotypic diversity in patients with identical compound heterozygous *PARK2* mutations, further emphasizing the role of other genetic and/or environmental factors in disease manifestation. It also further supports that loss of DaT expression precedes clinical manifestations of PD.

## Figures and Tables

**Figure 1 brainsci-07-00071-f001:**
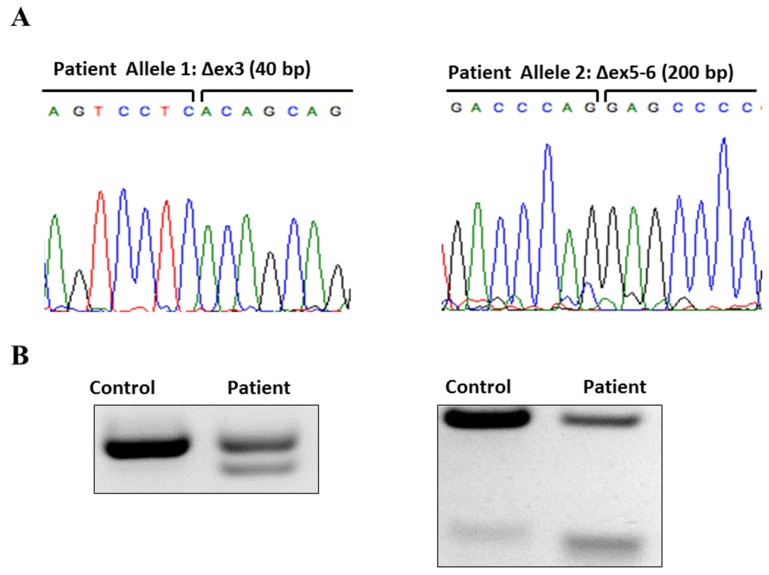
Panel (**A**) shows sequencing of patients’ cDNA generated by RT PCR with deletion in both alleles. Panel (**B**) shows confirmation of heterozygosity in each locus where a control subject has only wild type (normal) band in both analyzed loci; patients’ cDNA shows heterozygosity with wild type and smaller mutant bands. This was also previously published in a modified form as a part of Supplementary data, with the proband denoted “SM” and the younger brother denoted “PM” [18].

**Figure 2 brainsci-07-00071-f002:**
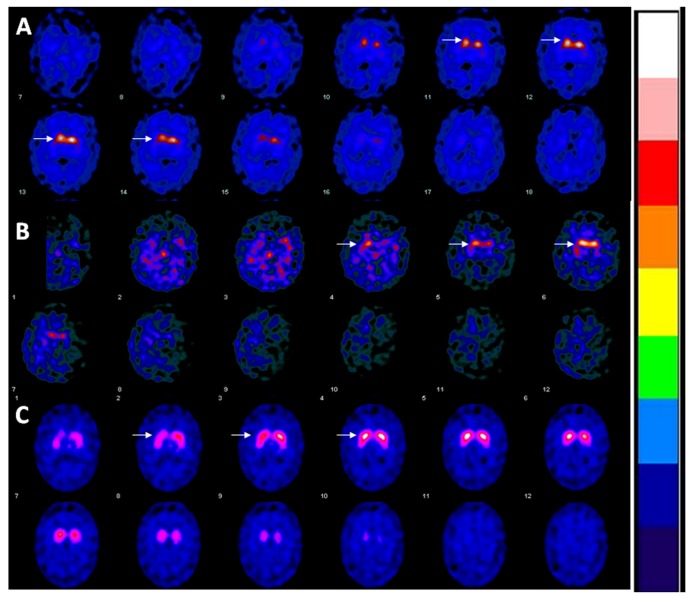
The younger sibling’s DaT-SPECT images (Rows (**A**)) shows asymmetrically decreased activity in the putamen bilaterally. The proband’s images (Rows (**B**)) show severely reduced uptake in the bilateral putamen with relative preservation in the caudate heads. (Rows (**C**)) are DaT-SPECT images from a matched control.

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
