# Peer review of "Phenotypic Discordance in Siblings with Identical Compound Heterozygous PARK2 Mutations"

_brainsci, 2017, doi:10.3390/brainsci7070071_

Round 1

Reviewer 1 Report

Phenotypic discordance in siblings with identical compound heterozygous PARK2 mutations.

Summary: The authors describe the clinical and phenotypic variation between two siblings with identical PARK2 mutations.  Interestingly, the brothers display remarkably different clinical phenotypes.  The older brother developed parkinsonism in his mid-20 and at 53 is wheel-chair bound while the younger developed exercise induced dystonia in his mid-30’s with no subsequent clinical progression.  Using DAT-SPECT, both brothers display abnormal dopamine uptake: the older brother has symmetric bilateral defects in the putamen, while the younger brother has asymmetric defects. This may represent the first report of abnormal DAT-SPECT without meeting PD criterion.

Comments:

This case study is very interesting and highlights a truly fascinating finding in that the younger brother remains largely asymptomatic despite decreases in dopamine uptake in the putamen.  The authors briefly speculate that environmental toxins could contribute to the differences in phenotype, but rule this out based on exposure.  As exercise may impact the progression of PD, could a difference in physical activity possibly account for the phenotypic differences?  Does the younger brother run with regularity, and if yes, for how long?  Did the brothers differ substantially in exercise prior to developing neurological symptoms? 

Author Response

The authors are not excluding an environmental risk factor contributing to the distinct phenotypes.  Rather, no clear environmental risk factors were evident by history.  Given the childhood age-of-onset of the proband and the fact that both brothers were raised in the same home (see Line 113), the authors hypothesize that genetic factors had a more influential role in the phenotypic discrepancy between the two brothers. 

In adulthood, the proband’s younger brother exercised 1-2 days per week on average, running several miles each time.  A sentence was added to relay this information (Line 90).

Reviewer 2 Report

This manuscript by Isaacs et al reported two siblings with confirmed identical compound heterozygous parkin mutations. Although the siblings had same Parkin mutations, they were phenotypically discordant. The analysis of phenotype/genotype correlation might reveal the pathomechanisms from the aspect of the association between genetic and environmental factors.

In this context, I am interested in this manuscript. I have, however, some issues with the discussion:

1. The author should show the family tree with the results of sanger sequences.

2. The decreasing serum levels and/or decreasing the consumption of caffeine might increase the risk of Parkinson’s disease, whereas elevating serum levels of urine might protect to the development of this disease. Therefore, the authors should state the caffeine consumption and serum urine levels.

3. The elder patient showed atypical PARK2 phenotype. Past investigations reported that some atypical PARK2 patients had Lewy bodies. Therefore, the authors should emphasis whether the both siblings had REM-sleep behavior disorders (should be assessed by RBD questionnaire) and hyposmia (should be assessed by UPSIT), which are alpha-synculeinopathy related biomarkers. Additionally, if possible, the authors should have demonstrated the 123I-MIBG cardiac scintigraphy of patients.

Author Response

1. “The author should show the family tree with the results of sanger sequences.“

A figure with the Sanger sequences for the brothers was added (see Figure 1).   Note that a modified form of Figure 1 was previously published in supplemental data of an article in Neurobiology of Disease (see Reference 18); the Figure has been altered to address the comments of Reviewer 2.  The brothers have no other siblings.  The Sanger sequences for the parents are not available.

2. “The decreasing serum levels and/or decreasing the consumption of caffeine might increase the risk of Parkinson’s disease, whereas elevating serum levels of urine might protect to the development of this disease. Therefore, the authors should state the caffeine consumption and serum urine levels.”

Caffeine consumption and uric acid levels have been added to the manuscript text (Lines 114-115 and Lines 113-114, respectively). 

3. “The elder patient showed atypical PARK2 phenotype. Past investigations reported that some atypical PARK2 patients had Lewy bodies. Therefore, the authors should emphasis whether the both siblings had REM-sleep behavior disorders (should be assessed by RBD questionnaire) and hyposmia (should be assessed by UPSIT), which are alpha-synculeinopathy related biomarkers. Additionally, if possible, the authors should have demonstrated the 123I-MIBG cardiac scintigraphy of patients.”

As now has been explicitly stated in the revised manuscript, neither brother exhibited RBD or hyposmia (Lines 74-75 and 93-94).  In addition, the proband had serial neuropsychological evaluations which demonstrated only mild executive dysfunction after almost 40 years of disease (Lines 86-87).  MIBG cardiac scintigraphy was not performed.
